# Use of vitamin and mineral supplements among adolescents living in Germany—Results from EsKiMo II

**DOI:** 10.3390/nu11061208

**Published:** 2019-05-28

**Authors:** Hanna Perlitz, Gert B.M. Mensink, Clarissa Lage Barbosa, Almut Richter, Anna-Kristin Brettschneider, Franziska Lehmann, Eleni Patelakis, Melanie Frank, Karoline Heide, Marjolein Haftenberger

**Affiliations:** Department of Epidemiology and Health Monitoring, Robert Koch Institute, 12101 Berlin, Germany; MensinkG@rki.de (G.B.M.M.); Lage-BarbosaC@rki.de (C.L.B.); RichterA@rki.de (A.R.); BrettschneiderA@rki.de (A.-K.B.); LehmannF@rki.de (F.L.); PatelakisE@rki.de (E.P.); Frank.Melanie@icloud.com (M.F.); Karoline.Heide@t-online.de (K.H.); HaftenbergerM@rki.de (M.H.)

**Keywords:** vitamin, mineral, dietary supplements, adolescents, EsKiMo

## Abstract

Dietary supplements may contribute to nutrient intake; however, actual data on dietary supplement use among adolescents living in Germany are rare. The aim of this analysis was to describe the current use of dietary supplements, its determinants, and reasons of use. Changes in supplement use over time were evaluated by comparing the results with those from EsKiMo I (2006). Data from the Eating Study as a KiGGS Module EsKiMo II (2015–2017) were used to analyze supplement intake according to sociodemographic, health characteristics, and physical exercise behavior of 12–17-year-olds (*n* = 1356). Supplement use during the past four weeks was assessed by a standardized computer assisted personal interview. Multivariable logistic regression was used to identify the association between supplement use and its determinants. Between 2015–2017, 16.4% (95%-CI: 13.0–19.7%) of the adolescents used dietary supplements, and its use decreased with lower levels of physical exercise and overweight. Most supplement users used only one supplement, often containing both vitamins and minerals. The most frequently supplemented nutrients were vitamin C and magnesium. The main reported reason to use supplements was ‘to improve health’. Prevalence of supplement use was slightly lower in 2015-2017 than in 2006 (18.5%; 95%-CI: 15.8–21.2%). The results underline the importance of including nutrient intake through dietary supplements in nutrition surveys.

## 1. Introduction

An optimal nutrient supply during the growth period of adolescence is important [1]. The majority of the population living in Germany has an adequate supply of almost all vitamins and minerals. Generally, with a balanced diet the requirements for essential nutrients will be met. However, for some nutrients, like in particular folate, iodine, and vitamin D, the nutrient status is suboptimal for large population groups in Germany. Accordingly, vitamin or mineral supplements are only recommended in Germany for medically diagnosed deficiencies or for vulnerable groups, such as infants, pregnant women, and the elderly. For adolescents, there are no recommendations for the preventive use of dietary supplements [2]. 

Nevertheless, sales for dietary supplements have increased in Germany and other western countries over the last decades [3,4]. The demand for supplements constituted a sale of 1.4 billion Euro in Germany in 2018 [4]. In light of the mostly adequate nutrient supply and the risks associated with an excessive intake of vitamins and nutrients, this trend should be observed critically [5].

Many studies described an increase in the use of dietary supplements, such as the National Health and Nutrition Examination Survey (NHANES) between 1971–2000 among adults [6]. The proportion of German adults who used dietary supplements during a period of seven days increased by six percent (12.3% to 18.1%) between 1997–1999 and 2008–2011 [7]. Up-to-date and representative information on supplement use among children and adolescents living in Germany is lacking. There are some studies on supplement use, but these are mainly regional and/or older studies and include different age groups. Among children and adolescents (2–18 years) who participated in the regional Dortmund Nutritional and Anthropometric Longitudinally Designed Study (DONALD) between 1986–2003, 7.5% reported the use of dietary supplements in a three-day-weighted dietary record [8]. 9.2% of children (9–12 years) from two German birth cohort studies (GINIplus and LISAplus) used dietary supplements (2005–2009) [9]. In the first representative German Eating Study as a KiGGS Module (EsKiMo) from 2006, one fifth of the adolescents (12–17 years) had used dietary supplements in the previous four weeks [10]. Current data about dietary supplement use and its determinants may help to estimate the risk of oversupply of micronutrients in specific groups. EsKiMo II, conducted from 2015 to 2017 by the Robert Koch Institute (RKI), provides recent data on the use of dietary supplements. The present analysis aims to quantify the use of vitamin and mineral supplements in association with some determinants, and to evaluate the reasons for dietary supplement use among adolescents living in Germany. A major advantage of the current analysis is the possibility to describe the change in the prevalence of dietary supplement use between 2006 and 2015–2017 by comparing results from EsKiMo I and EsKiMo II. 

## 2. Methods

### 2.1. Study Design and Study Population

EsKiMo II was conducted from June 2015 until September 2017 as part of the second wave of the German Health Interview and Examination Survey for Children and Adolescents (KiGGS Wave 2). The aim of the cross-sectional EsKiMo II was to assess the dietary behavior of children and adolescents and to identify changes in food consumption in the last decade by comparison with the previous study EsKiMo I, conducted in 2006. The EsKiMo II study protocol was consented with the Federal Commissioners for Data Protection and approved by the Hannover Medical School ethics committee in June 2015 (Number 2275–2015). Written informed consent was obtained from all parents or legal guardians and participants aged 14 years and older prior to the study interviews and examinations. Details on the methodology of EsKiMo II can be found elsewhere [11,12,13]. 

Altogether, 2644 children and adolescents aged 6–17 years who took part in KiGGS Wave 2 participated in EsKiMo II (participation rate 59.4%). The current analysis is limited to 1356 adolescents aged 12–17 years, as supplement use for this age group was assessed in the same way as in EsKiMo I, which allows describing changes in supplement use between 2006 and 2015–2017. Data of 1272 adolescents who participated in EsKiMo I was included for trend analysis of vitamin and mineral supplement use over time. 

### 2.2. Assessment of Supplement Use

Dietary supplement use was assessed using a standardized computer assisted interview within the Dietary Interview Software for Health Examination Studies DISHES [14] by trained nutritionists during home visits. The use of dietary supplements was ascertained by the following question: “Have you taken dietary supplements (like vitamins or minerals) in form of tablets, drops etc. in the last four weeks?" In case of a positive response, types (name and brand), frequencies of use, dosage form, and amounts of the supplement and reasons for using were asked. Dietary supplements were selected from a database, which was integrated in the software. This supplement database is an update from previous dietary surveys conducted in Germany by the RKI and the Max Rubner Institute. Supplements which were not included in the database were recorded as a free text with the name, brand, and dosage form. If available, a photo of the supplement packaging was taken. Afterwards, nutrient composition of all recorded supplements was checked with the information from the package, from the internet and/or from the manufacturer, and, if necessary, updated or added in the database. Reasons for supplement use were asked by predefined categories (‘to improve health’, ‘based on a doctor’s recommendation’, ‘increase of physical and mental performance’, ‘read/heard about beneficial information’, ‘compensation of low fruit/vegetables consumption’, ‘based on a pharmacist’s recommendation’), with the possibility of free texts for other reasons (more than one answer was possible). 

The current analysis is limited to dietary supplements containing vitamins or minerals and considers both freely available and medically prescribed dietary supplements. Supplements containing neither vitamins nor minerals or containing vitamins or minerals in homeopathic doses were assigned to the category ‘other dietary supplements’ and excluded from this analysis. Protein and dietetic products were assigned to foods and not considered in this analysis. Vitamin and mineral supplements were categorized as: vitamins, minerals, and combined preparations containing both vitamin/s and mineral/s. 

### 2.3. Assessment of Other Variables

EsKiMo II participants were visited about three to six months after participation in KiGGS Wave 2. Sociodemographic, lifestyle, and health characteristics were assessed within KiGGS Wave 2 by self-administered questionnaires completed by the parents and by the adolescents aged 11 years old and older themselves [15]. Socio-economic status (SES) was based on information about education level, occupational status, and net household income of the parents and categorized into low, medium, and high SES [16]. Attended school types were categorized as lower secondary school, upper secondary school, and other school types. Participants were defined as having a migration background, when they or at least one parent were not born in Germany or did not have the German nationality. Three categories for residence region were constructed according to federal states: north (Schleswig-Holstein, Hamburg, Lower Saxony, Bremen, Berlin, Brandenburg, Mecklenburg-Western Pomerania), middle (North Rhine-Westphalia, Hesse, Saxony, Saxony-Anhalt, Thuringia), and south (Rhineland-Palatinate, Baden-Wuerttemberg, Bavaria, Saarland). Community size was categorized as: <5000 inhabitants, 5000 to 20,000 inhabitants, 20,000 to under 100,000 inhabitants, and >100,000 inhabitants. During the personal interviews in EsKiMo II, questions about the body height and weight of the adolescents were asked. Based on this self-reported information, body mass index (BMI) (body weight in kg/body size in m²) was calculated and assigned into age- and sex-specific BMI categories according to Kromeyer-Hauschild (underweight, normal weight, overweight) [17,18]. Furthermore, subjective health status was rated by adolescents of 11 years and older themselves in four categories, ranging from excellent to poor. Physical exercise as a specific and more intensive type of physical activity includes all kinds of sports, but without physical education at school. Questions about the usual duration (hours/minutes per week) of physical exercise were asked and replies were categorized into ‘low’ (less than one hour /week), ‘medium’ (1–3 h/week) and ‘high’ (more than 3 h/week). 

### 2.4. Statistical Analysis

Prevalence of supplement use is presented according to sociodemographic and other characteristics. In addition, multivariate logistic regression was conducted to analyze the independent associations of these sociodemographic and other characteristics with supplement use. The associations were adjusted for all other variables. For supplement users, the type, number, and frequency as well as the reasons for vitamin or mineral supplement use are described. Finally, changes in the prevalence of dietary supplement use between 2006 and 2015–2017 are examined. All analyses were performed with SAS Version 9.4 (SAS Institute, Cary, NC, USA). The criterion for statistical significance was set at *p*-value< 0.05. A weighting factor was applied to correct for deviations from the population structure according to age (in years), sex, region (as of 31.12.2015), nationality (as of 31.12.2014), and education level of the parents (Mikrozensus 2013), as well as to consider differences in participation´s probability according to seasonality, SES of the family, and school type. In order to take the clustered design into account (with a stronger correlation of the participants within a community compared to a totally random group), the SAS survey procedures were applied. Data of 1267 adolescents from EsKiMo I are included for trend analysis. Study procedures and instruments of EsKiMo I are generally the same as for EsKiMo II and are described elsewhere [19]. Prior analyses of EsKiMo I were calculated using a weighting factor to correct for the disproportional higher number of participants from the Eastern part of Germany as well as deviations in age, sex, and nationality from the general population [10]. For the present analysis, these prevalence estimates were recalculated with a weighting factor constructed as described above and correcting deviations from the population structure of 2004. For comparison of prevalence estimates from EsKiMo I and EsKiMo II taking into account demographic changes over time, the EsKiMo I prevalence estimates were standardized to the sex- and age-structure of the population underlying the EsKiMo II data.

## 3. Results

In total, 16.4% of the adolescents (girls: 18.8%, boys: 14.0%) aged 12 to 17 years had used vitamin or mineral supplements in the previous four weeks (Table 1). The proportion of supplement use was similar across age groups, SES, type of school, migration background, region of residence, and community size (Table 1). Additionally, there was no association between self-assessed health status and dietary supplement use (data not shown).

The multivariable logistic regression showed that sex, weight status, and physical exercise were independent determinants of dietary supplement use. Boys use dietary supplements less frequently than girls (OR: 0.60 (0.38–0.94) and adolescents with overweight use dietary supplements less frequently compared to adolescents with normal weight (OR: 0.41 (0.21–0.79) (Table 1). The use of dietary supplements was lower for adolescents with low levels of physical exercise (OR: 0.56 (0.33–0.95) compared to those with a high level of physical exercise (Table 1). 

Among the dietary supplement users, 36.9% utilize vitamin supplements, 40.8% mineral supplements, and 46.4% a combination of both vitamins and minerals (Table 2), with no differences regarding sex (data not shown). During the previous four weeks, the majority of the users had consumed only one kind of dietary supplement (72.7%), and about a quarter (27.3%) had consumed more than one (Table 2). Around 28% of the vitamin and mineral supplements were used daily (6–7 times a week) (data not shown). The most commonly used vitamin supplements contained vitamin C (43.9%), followed by vitamin D (41.1%) and vitamin B12 (30.4%). The reported mineral supplements most often contained magnesium (45.9%), zinc (28.1%), and iron (24.1%; Table 2).

The most common reason for using vitamin and mineral supplements during the last four weeks was ‘to improve health’ (59.3%). One fifth of the participants (20.7%) reported to use supplements ‘based on a doctor’s recommendation’, followed by 17.7% of the adolescents who reported an ‘increase of physical and mental performance’ as motivation. Further, less frequently reported reasons for using supplements were ‘read/heard about beneficial information‘ (7.2%), ‘compensation for low fruit/vegetables consumption’ (7.2%), ‘based on a pharmacist’s recommendation’ (3.6%). Among other reasons assessed as free text, the most often answer was ‘based on KiGGS examination results’ (2%) (Figure 1). 

The prevalence of dietary supplement use in the previous four weeks decreased slightly, but not statistically significantly, from 18.5% to 16.4% between Eskimo I (2006) and Eskimo II (2015–2017) (Table 3). 

## 4. Discussion

To our knowledge, no nationwide overview of recent dietary supplement use among adolescents has been presented for Germany since EsKiMo I in 2006. Our analysis show that 16.4% of the adolescents aged 12 to 17 years had used vitamin or mineral supplements in the last four weeks. 

In EsKiMo II, supplement use is associated with sex, weight status, and physical exercise. Dietary supplement use was similar across age groups, SES, type of school, migration background, region of residence, community size, and subjective health status.

Comparison of our study results with other studies is difficult due to differences in methods of data collection, definitions of dietary supplements, timeframes, and age groups. In most western countries, the use of dietary supplements among children and adolescents is higher than in Germany [20,21,22,23], which is also confirmed for adults [24]. A third of the participants aged 9–18 years of the American dietary survey NHANES (2011–2014) reported dietary supplement use during the last month [20]. In an Australian study (2014–2015), 20.1% of the adolescents (10–17 years old) had used dietary supplements during the past two weeks [21]. 

Previous studies including children and adolescents showed inconsistent findings considering sex differences in dietary supplement use. Many studies observed no significant differences in dietary supplement use by sex [21,22,25]. Parents probably determine the supplement use for their children, which may largely explain the absence of sex differences. The German DONALD study showed a higher prevalence of dietary supplement use among boys [26]. A Polish study observed that supplement use was more common among girls [23]. Other studies observed higher supplement use among children and adolescents with underweight, which is similar to our observations [25,27,28]. Previous findings regarding the effect of physical exercise on dietary supplement use are heterogeneous and only partly comparable due to differences in data assessment and definition of physical exercise. In EsKiMo II, adolescents who reported less than one hour of physical exercise per week used less frequently dietary supplements. This finding is consistent with the results from EsKiMo I and NHANES, although different definitions for physical exercise were applied [10,27]. 

Previous studies were also inconsistent with regard to the effect of age on dietary supplement use. NHANES and a Korean study found a higher prevalence of dietary supplement use for younger age groups [20,22]. However, unlike our study, these studies also included children under the age of 12 years. Other studies observed a higher prevalence of supplement use for older age groups (15–18 years; 16–18 years) [8,23]. In EsKiMo II, dietary supplement use is similar regarding SES. Previous studies consistently showed that a higher prevalence of dietary supplement use was associated with higher SES, higher education of the parents or higher household income [21,22,25,26,29]. For EsKiMo I, supplement use significantly differed by school type [10], whereas such differences were not significant for EsKiMo II, but the categories are not exactly the same. EsKiMo II did not observe differences in supplement use by migration status. The definition of migration status is very different between studies and countries, making it difficult to compare results. Although a Polish study with a small sample size observed a higher supplement use among adolescents living in bigger cities [23], dietary supplement use among adolescents was similar through different community sizes in EsKiMo II. This is similar to the results of the German Health Interview and Examination Survey (DEGS1; 2008–2011), which also observed no statistically significant differences in supplement use according to the community size and region of residence among German adults [7]. 

In EsKiMo II, the most frequently dietary supplements used were a combination of vitamins and minerals. Vitamin C and magnesium were the most supplemented micronutrients, similarly to results of an Australian study [21]. Other studies showed related results: in NHANES and DONALD, the most frequently dietary supplement used was also vitamin C, but the most frequent mineral supplemented was calcium and not magnesium [26,30]. A German consumer survey with adults observed similar results to EsKiMo II, with even higher prevalences for vitamin C (53%) and magnesium (59%) [31]. Vitamin C and magnesium are also the most commonly supplemented micronutrients by adults [24,31,32,33]. However, there are no recommendations and no need for adolescents to supplement vitamin C and magnesium [2,34].

Reasons for dietary supplements use can be diverse. Within EsKiMo II, the most commonly reported reason for vitamin or mineral supplements use was ‘to improve health’ (59.3%), which was also identified as the most frequent motive in previous studies (CRN Consumer survey: 58%; NHANES 2011–2014: 38.0%) [30,35]. The German NEMONIT study, for example, indicated ‘prevention of nutrient deficiencies (or covering increased nutrient requirements)’ (62.4%) and ‘achievement or improvement of general well-being’ (34.7%) as the most common motives for dietary supplement use among German adults [36]. For the participants of EsKiMo II, the recommendations of a physician were reported as the second most important reason for dietary supplements use, but this only applies to one in five adolescents (20.7%). 2% of the supplement users indicated using dietary supplements as a consequence of the personal evaluation they received after taking part on KiGGS Wave 2, from which the EsKiMo II study sample was drawn. 

The prevalence of dietary supplement intake decreased slightly, but not significantly, between EsKiMo I (2006) and EsKiMo II (2015–2017). Data from the longitudinal DONALD-study among children and adolescents aged 2–18 in Germany were used to describe a time trend in dietary supplement use from 1986–2003. Supplement use peaked between 1994 and 1996 [26]. One study among adolescents in the US showed similar prevalences between 2003–2004 and 2013–2014 [37]. While former studies for adults reported an increase in dietary supplement intake [6,38], more recent results showed a stagnation in its intake and a declining trend for some particular dietary supplements [39,40]. The observed stagnation may seem surprising, considering the many lifestyle changes detected in the last decades. However, in Germany, the supply of most vitamins and minerals through natural sources meets the recommended level and this situation has not changed substantially during the last ten years. 

Currently, there are no recommendations for the use of vitamin and mineral supplements among adolescents in Germany. The German Nutrition Society recommends a balanced diet to meet the requirements for essential nutrients. The use of dietary supplements is only recommended for individuals with certain medically diagnosed diseases [1]. Nevertheless, a deficit for some vitamins, such as vitamin D, folic acid, and iodine has been reported among adolescents [41]. In addition, there is a certain risk of overconsumption, in particular when more than one dose of supplements or several different supplements containing the same nutrients are consumed every day in combination with fortified foods. For the latter, the consumer is often unaware of the specific fortifications. The growing market for both product groups, including a less controllable sale and distribution over the internet, may increase the risk of side effects. Therefore, in Germany, the Federal Institute for Risk Assessment defined a recommendation for maximum levels of the amount of vitamins and minerals contained in dietary supplements [5]. Furthermore, within the observed time period, results of large randomized controlled trials could not confirm the expected additional benefits of a regular supplement intake, e.g. for vitamin D. This may have counteracted marketing and promoting activities, therefore slowing down further increases in supplement intake.

Among the strengths of our study are the representative study population and the assessment through standardized personal interviews, which took place mainly at the participant’s home. Data collection of vitamin and mineral supplements was carried out using a comprehensive supplement database. For more detailed information, pictures of the dietary supplement packages were taken. The nutrient composition of all dietary supplements reported was checked. The cross-sectional design of the study can represent a limitation for certain research questions, since no direct causalities can be derived from the observed correlations. In addition, selection bias is possible due to the previous participation of the adolescents in KiGGS Wave 2. To minimize the effect of selective participation, population weights were applied to correct deviations from the German population. A further limitation could be the use of self-reported body height and weight to estimate weight status. Self-reported body weight and height of a subgroup of participants who had also taken part in physical examinations of KiGGS Wave 2 were individually compared with the measured data from KiGGS Wave 2. This comparison showed only small differences and a high correlation between the measured and self-reported data.

## 5. Conclusions

Almost one fifth of adolescents reported to use dietary supplements within the last four weeks; most of them took one dietary supplement. Vitamin C and magnesium were the most commonly supplemented nutrients, although predominantly no nutrient deficiency exists for these nutrients. Nevertheless, critical nutrients, such as vitamin B12 and vitamin D, were also taken frequently [34,41]. Boys, adolescents with lower physical exercise levels and with overweight were less likely to use dietary supplements. The considerable use of dietary supplements among adolescents underlines the importance of considering dietary supplements intake into account for assessing nutrient intake in nutrition surveys. Furthermore, possible oversupply should be monitored and information on the risk and benefits of specific dietary supplements should be provided to the population.

## Figures and Tables

**Figure 1 nutrients-11-01208-f001:**
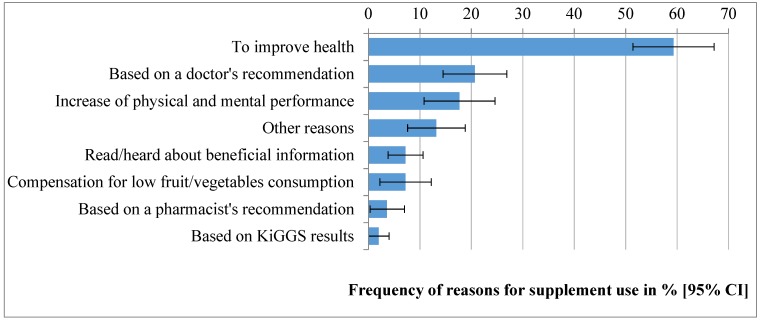
Prevalence and 95%-confidence intervals (CI) of reasons for dietary supplement use among adolescents (12–17 years) in EsKiMo II (2015–2017), *n* = 228 (weighted for the German population of 2015).

**Table 1 nutrients-11-01208-t001:** Associations between vitamin and/or mineral supplement use among adolescents (12–17 years) and determinants (sociodemographic characteristics, weight status, and physical exercise) in EsKiMo II (2015–2017), *n* = 1356.

Vitamin and/or Mineral Supplement Use	Prevalence ^1^	Multivariate Logistic Regression Analysis ^1^
	*n* = 1356	*n* = 1223
	% (95% CI)	adjusted OR (95% CI) ^2^
**Total**	16.4 (13.0–19.7)	-
**Sex**		
Girls	18.8 (14.5–23.2)	Ref.
Boys	14.0 (9.9–18.1)	0.60 (0.38–0.94) *
**Age group**		
12–13 years	14.4 (10.3–18.4)	0.76 (0.46–1.26)
14–15 years	19.4 (13.2–25.5)	1.30 (0.77–2.20)
16–17 years	15.2 (10.4–20.0)	Ref.
**Socio-economic status (SES) ^3^**		
Low	12.3 (5.4–19.2)	0.63 (0.26–1.53
Medium	15.3 (11.7–19.0)	0.66 (0.42–1.05)
High	22.3 (16.2–28.3)	Ref
**Type of school ^4^**		
Lower secondary school	14.5 (10.9–18.1)	0.84 (0.53–1.32)
Upper secondary school	19.2 (14.4–24.0)	Ref.
Other school types	13.4 (4.8–22.0)	1.02 (0.40–2.59)
**Migration background ^5^**		
Yes	15.4 (6.9–23.9)	0.89 (0.48–1.67)
No	16.5 (13.2–19.9)	Ref.
**Region of residence**		
North	13.8 (8.5–19.0)	0.74 (0.40–1.39)
Middle	16.7 (11.7–21.6)	1.11 (0.62–1.96)
South	17.7 (11.2–24.3)	Ref.
**Community size**		
<5000 inhabitants	15.0 (8.5–21.5)	Ref.
5000–<20,000 inhabitants	19.9 (11.6–28.2)	1.04 (0.52–2.09)
20000–<100,000 inhabitants	16.9 (10.9–22.9)	0.83 (0.39–1.74)
≥100,000 inhabitants	13.4 (8.2–18.6)	0.76 (0.36–1.63)
**Weight status**		
Underweight	20.7 (11.2–30.1)	1.33 (0.73–2.41)
Normal weight	17.0 (13.1–21.0)	Ref
Overweight	9.3 (3.9–14.7)	0.41 (0.21–0.79) *
**Physical Exercise**		
<1 h/week	11.5 (7.2–15.9)	0.56 (0.33–0.95) *
1–3 h/week	14.8 (10.9–18.8)	0.73 (0.47–1.12)
>3 h/week	19.9 (15.0–24.9)	Ref.

^1^ weighted for the German population of 2015; ^2^ adjusted for all other variables; ^3^ Socio-economic status: *n* (missing) = 19; ^4^ Type of school *n* (missing) = 62, including participants who already finished school; ^5^ migration background *n* (missing) = 9; *n* = number of subjects; OR = odds ratio; CI = confidence interval; * OR is statistical significantly different from the reference with *p*-value < 0.05.

**Table 2 nutrients-11-01208-t002:** Frequency ^1^ of the number and type of dietary supplements used among adolescents (12–17 years) in EsKiMo II (2015–2017), *n* = 1356.

	Total	Supplement User
	*n* = 1356	*n* = 234
	% (95% CI)	% (95% CI)
**Type of supplement ^2^**		
Vitamin/s	6.0 (4.3–7.7)	36.9 (29.4–44.5)
Mineral/s	6.7 (4.5–8.8)	40.8 (31.9–49.8)
Combination of vitamin/s and mineral/s	7.6 (5.5–9.7)	46.4 (38.8–54.1)
**Number of supplements**		
1 supplement	11.9 (9.3–14.4)	72.7 (64.8–80.6)
>1 supplement	4.5 (2.8–6.2)	27.3 (19.4–35.2)
**Vitamins ^1^**		
Vitamin A	1.2 (0.6–1.9)	7.5 (3.7–11.9)
Thiamin	4.2 (2.7–5.7)	25.6 (17.3–34.0)
Riboflavin	4.0 (2.5–5.5)	24.2 (16.0–32.4)
Niacin	3.6 (2.1–5.1)	22.3 (14.3–30.3)
Pantothenic acid	3.3 (2.0–4.6)	20.5 (13.7–27.7)
Vitamin B6	4.2 (2.8–5.5)	25.7 (18.1–33.3)
Biotin	3.6 (2.3–5.0)	22.2 (15.7–28.6)
Folate	4.1 (2.6–5.6)	25.2 (17.6–32.8)
Vitamin B12	5.0 (3.3–6.7)	30.4 (22.1–38.7)
Vitamin C	7.2 (5.4–9.0)	43.9 (35.8–52.1)
Vitamin D	6.7 (4.6–8.8)	41.1 (32.6–49.5)
Vitamin E	3.6 (2.3–5.0)	22.3 (15.1–29.6)
Vitamin K	0.9 (0.3–1.6)	5.7 (2.3–9.2)
**Minerals ^1^**		
Calcium	3.2 (1.6–4.9)	19.8 (11.9–27.7)
Copper	1.5 (0.5–2.5)	8.9 (3.1–14.6)
Fluoride	0.4 (0.0–0.9)	2.3 (0.0–5.4)
Iron	3.9 (2.2–5.7)	24.1 (15.1–33.0)
Iodine	1.4 (0.4–2.4)	8.5 (2.7–14.4)
Potassium	0.3 (0.1–0.5)	1.9 (0.5–3.4)
Magnesium	7.5 (5.1–9.9)	45.9 (36.9–54.9)
Manganese	1.3 (0.4–2.2)	17.5 (2.6–13.0)
Molybdenum	1.3 (0.4–2.3)	8.2 (2.6–14.0)
Sodium	0.1 (0.0–0.2)	0.7 (0.0–1.5)
Phosphorus	0.4 (0.0–0.9)	2.7 (0.2–5.2)
Selenium	1.9 (0.8–2.9)	11.5 (5.6–17.4)
Zinc	4.6 (2.9–6.3)	28.1 (19.1–36.4)

^1^ weighted for the German population of 2015; ^2^ due to multiple supplement use and multiple active components the sum of the prevalences by type of supplements or active components may deviate from the prevalence of total supplement use as displayed in Table 1; CI = confidence interval.

**Table 3 nutrients-11-01208-t003:** Trend analysis of dietary supplement use among adolescents (12–17 years) between EsKiMo I (2006) and EsKiMo II (2015–2017).

	EsKiMo I	EsKiMo I	EsKiMo II
	*n* = 1267	*n* = 1267	*n* = 1356
	% (95% CI)	% (95% CI)	% (95% CI)
	Weighted for 2004	Weighted for 2015	Weigthed for 2015
**Total**	18.5 (15.8–21.2)	18.5 (15.8–21.2)	16.4 (13.0–19.7)
**Girls**	19.3 (15.3–23.4)	19.4 (15.3–23.5)	18.8 (14.5–23.2)
**Boys**	17.7 (14.3–21.1)	17.7 (14.3–21.1)	14.0 (9.9–18.1)
**Type of supplement** ^1^			
Vitamin/s	6.2 (4.7–7.8)	6.2 (4.7–7.8)	6.0 (4.3–7.7)
Mineral/s	9.9 (7.6–12.2)	9.9 (7.6–12.3)	6.7 (4.5–8.8)
Combination of vitamin/s and mineral/s	6.1 (4.4–7.8)	6.1 (4.4–7.8)	7.6 (5.5–9.7)
**Number of supplements**			
1 supplement	13.6 (11.1–16.1)	13.7 (11.2–16.2)	11.9 (9.3–14.4)
>1 supplement	4.9 (3.2–6.5)	4.9 (3.2–6.5)	4.5 (2.8–6.2)

^1^ due to multiple dietary supplement use and multiple active components, the sum of the prevalences by type of dietary supplements or active components may deviate from the prevalence of total dietary supplement use; CI = confidence interval.

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
