# Peer review of "Use of vitamin and mineral supplements among adolescents living in Germany—Results from EsKiMo II"

_nutrients, 2019, doi:10.3390/nu11061208_

Round 1

Reviewer 1 Report

This paper is very straight-forward and easy to understand. A few minor comments:

L16: “changes in its intake” – it is unclear what changes you are referring to. The comparison with 2006 data is mentioned in L21 but would be better suited to following the “changes in its intake” statement on L16.

L20-21: It is not clear what associations you were evaluating.

L42: Does reference 3 refer to U.S. revenue from production of vitamins & nutritional supplements rather than German data?

L252: Are there any supplementation recommendations for this population group within Germany? The discussion could introduce more context regarding public healthy policy and legislative framework.

L280: The use of self-reported weight and height, and subsequent categorisation of body weight status, should be acknowledged as an important limitation of this study, and caution should be urged when interpreting the findings of different rates of supplementation across body weight status.

Author Response

Thank you for the helpful comments. The answers are detailed in the word document below.

Reviewer 2 Report

The Use of vitamin and mineral supplements among German population is an interesting and a relevant study as it presents data since the last survey. But the study seems slightly repetitive to the other studies done on the similar subject. I would recommend the authors to highlight the differences in their study to stand out from several other studies done on the subject. 

Additionally, I feel it would be helpful to the manuscript if the authors could add any data or could comment on the negative effects on the intake of these extrogenous vitamins and minerals. 

The introduction seemed a little confusing and did not have enough details for the reader to understand the material of the paper (particularly the significance and outcomes of previous surveys). I would recommend the authors to clarify and include more details in the introduction.  

Lastly I would also recommend the authors to elaborate the results. I think it is important for the authors to explain the reasons for the absence of a significant change in the results of Eskimo I and Eskimo II, particularly considering a major life style change since 2006. 

On the positive note, the collection and extensive categorization of the data  and identifying the intake of each vitamin was very impressive. 

Author Response

Thank you for the helpful comments. The answers are formulated in the word document.
